# The international trial of nasal oxygen therapy after cardiac surgery (NOTACS) in patients at high risk of postoperative pulmonary complications: Economic evaluation protocol and analysis plan

Siddesh Shetty[1], Melissa Duckworth[2], Richard Norman[3], Jacquita Affandi[3,4], Sarah Dawson[2,5], Julia Fox-Rushby[1]*, On behalf of NOTACS investigators[¶], On behalf of NOTACS Trial Steering Committee[¶], On behalf of NOTACS DMEC Committee[¶]

1 School of Life Course and Population Sciences, King's College London, London, United Kingdom, 2 Papworth Trials Unit Collaboration, Royal Papworth Hospital NHS Foundation Trust, Cambridge, United Kingdom, 3 Curtin School of Population Health, Curtin University, Perth, Western Australia, 4 Fiona Stanley Hospital, Perth, Western Australia, 5 MRC Biostatistics Unit, School of Clinical Medicine, University of Cambridge, Cambridge, United Kingdom

¶ Membership of the NOTACS investigators, NOTACS Trial Steering Committee and NOTACS DMEC Committee is provided in the S2 Appendix.
* julia.fox-rushby@kcl.ac.uk

## Abstract

### Introduction

High-Flow Nasal Therapy (HFNT) is an innovative non-invasive form of respiratory support. Compared to standard oxygen therapy (SOT), there is an equipoise regarding the effect of HFNT on patient-centred outcomes among those at high risk of developing postoperative pulmonary complications after undergoing cardiac surgery. The NOTACS trial aims to determine the clinical and cost-effectiveness of HFNT compared to SOT within 90 days of surgery in the United Kingdom, Australia, and New Zealand. This protocol describes the methods and analyses planned for economic evaluation embedded within the ongoing NOTACS trial.

### Methods and analysis

The economic evaluation will identify, measure and value resources and health outcomes in both trial arms and compare changes in costs with 'days alive and at home' and EQ-5D-5L quality adjusted life years (QALYs) from the perspective most relevant to the decision-making country. Results from pooling data across the trial will use health and social care sector perspective. All patient-specific data including hospital/community care and health outcomes will be collected prospectively. Unit costs will be sourced from national, published or local data. Missing data will be assessed, with values replaced depending on assumed mechanism of missingness, and impact of replacement on cost-effectiveness assessed. Costs and outcomes by trial arm will be presented as components and totals per patient using a range of descriptive statistics. Regression models for costs and effects will account

**Data Availability Statement:** The data that support the findings of this study will be made available following journal and funder requirements. Details of data accessibility will be included when the results of the study are published.

**Funding:** This work in the United Kingdom was supported by funding from the National Institute for Health Research (NIHR) Health Technology Assessment, grant number: NIHR128351 (AK (PI), JFR, SV, GK, GJM, GM, ME, CF, JS, YC, VZ, VB, RMc, GB). The views expressed are those of the authors and not necessarily those of the NIHR or the Department of Health and Social Care, UK. The Australian recruitment was funded by Medical Research Future Fund International Clinical Trial Collaboration, grant number: MRF2006100 (EL (PI), JC, AK, RP, AM, CR, DP, YD, SY, MR) and by Green Lane Research and Educational Fund in Aotearoa New Zealand, grant numbers: 24/23/4167 and 21/23/4159 (RP, SMcG). Fisher and Paykel (F&P) loaned the HFNT equipment to the sites and provided education and training on its use and disposables required. Neither the funders nor F&P had any role in study design, data collection and analysis, decision to publish, or preparation of the manuscript.

**Competing interests:** The authors have declared that no competing interests exist.

for patient characteristics, quality of life and health service utilization at baseline. Uncertainty in parameters, sampling and heterogeneity will be addressed through deterministic, probabilistic and subgroup analyses to assess the impact of varying methods and assumptions for costs, outcomes and approaches used in base-case analysis. Results will be interpreted using recommended national cost-effectiveness thresholds.

## Registration details

The study is registered with ISRCTN (ISRCTN14092678) on 13/05/2020. ISRCTN is a primary registry of the WHO ICTRP network and includes all items from the WHO Trial Registration data set.

## Introduction

United Kingdom (UK), Australia (AUS), and New Zealand (NZ) reported a total of 42,456 cardiac surgical procedures in 2022–23 [1, 2]. Patients undergoing cardiac surgery are at risk of developing postoperative pulmonary complications (PPC), leading to prolonged hospitalisation and an increase in healthcare costs [3, 4]. The risk is further exacerbated in patients having intrinsic respiratory disease and lower airway obstruction, obese patients, and heavy smokers [5]. The peri-extubation period is crucial in managing such high-risk patients. These patients more often require prolonged ventilatory support and readmission during recovery [6, 7].

High-Flow Nasal Therapy (HFNT) is an innovative non-invasive form of respiratory support that has increasingly gained attention as a bridge between low flow oxygen and continuous positive airway pressure (CPAP) [8]. HFNT has physiological benefits and is better tolerated by patients compared to other techniques of non-invasive ventilation (NIV) and conventional oxygen therapy [9]. Recent systematic reviews have ascertained the efficacy and safety of HFNT in adult post-operative patients by demonstrating a reduction in length of hospital stay and the need for respiratory support [10, 11]. There is however an equipoise regarding the use of HFNT in patients at high risk of developing PPC undergoing cardiac surgery.

The NOTACS trial is an adaptive, multi-centre, multi-country, parallel group, randomised controlled trial (RCT) with embedded economic evaluation. The trial compares the use of HFNT versus SOT in patients at high risk of developing PPC after undergoing cardiac surgery. The trial has two primary aims; to determine whether prophylactic use of HFNT for a minimum of 16 hours after tracheal extubation increases days alive and at home (DAH) at 90 days after surgery, and to estimate the incremental cost-effectiveness and cost-utility of HFNT versus SOT at 90 days after surgery.

There is very little evidence available on the cost-effectiveness of HFNT. A decision model from England found HFNT to be cost saving in ICU patients but indicated limitations with clinical effectiveness and costing methods [12]. Similarly, a US-based model found HFNT to be cost-effective in COPD patients on long-term oxygen but is based on findings from a single-centre trial that used outcomes from a different healthcare setting [13]. The limited available economic evidence is mostly based on decision models, heterogenous population subgroups, different settings that have limitations for input costs, consequences or are funded by manufacturers [12–16]. Therefore, the actual cost-effectiveness of HFNT in adult patients at high risk of developing PPC undergoing cardiac surgery is currently unknown.

The economic evaluation alongside NOTACS is designed to provide the first international economic evidence on the use of HFNT for adults at high risk of developing PPC undergoing

cardiac surgery. The primary objective is to assess the cost-effectiveness (i.e. incremental cost per gain in 'days alive and at home' at 90 days after surgery (DAH90)) and cost-utility (i.e. incremental cost per Quality Adjusted Life Years (QALY) gained) for HFNT versus SOT comparison from the health and social care sector perspective. The secondary objectives include addition of patient/family perspective and estimation of cost-effectiveness at 30 days after surgery. This protocol paper (Version 1.0, 23rd September 2024) describes the methods and analyses planned for this economic evaluation.

## Methods and analysis

### Ethics

The study received ethics approval in the United Kingdom from Yorkshire & The Humber-Leeds West Research Ethics Committee, Newcastle upon Tyne, UK on 10th June 2020 (REC ref: 20/YH/0133) and HRA and Health and Care Research Wales on 19th June 2020. In Australia, the approval was obtained from South Metropolitan Health Service Human Research Ethics Committee, MURDOCH, Western Australia on 3rd September 2021 (ref: RGS0000004935) and 19th May 2022 (ref: RGS0000005326), from St John of God Health Care Human Research Ethics Committee (ref: 1870) on 13th October 2021, a reciprocal ethics approval from Curtin University Human Research Ethics Committee on 12th November 2021 (ref: HRE2021-0719) and West Australian Aboriginal Health Ethics Committee (ref: HREC1146) on 29th April 2022. In New Zealand, the approval was obtained from the Southern Health and Disability Ethics Committee, New Zealand on 12th October 2021 (ref: 21/STH/213).

### Study design

The NOTACS trial protocol (Trial registration ID: ISRCTN14092678) and statistical analysis plan are published elsewhere following the SPIRIT recommendations (Fig 1 and S1 Table in S1 Appendix) [17–19]. The trial examines patient centred outcomes for HFNT versus SOT comparison by using a single-blind strategy that assigns eligible patients randomly in a 1:1 allocation ratio, until a sample size of 1280 cases is achieved by recruiting participants from 10 sites in the UK, 7 in AUS and 1 in NZ. The trial population includes adult patients undergoing elective or urgent first-time or redo cardiac surgery performed on cardiopulmonary bypass, having one or more clinical risk factors for PPC. Randomisation is performed after surgery initiation and patients are followed up for 90 days after surgery.

### Intervention and control arms

After cardiac surgery, patients are transferred to intensive care units (ICU). Once a patient fulfils the local hospital criteria for extubation, they are extubated as per the trial extubation protocol [17]. Those randomised to the intervention arm receive HFNT for a minimum of 16 hours with up to one hour off treatment allowed for hospital moves or physiotherapy mobilisation. Following the extubation protocol, patients allocated HFNT are started off on 40% fraction of inspired oxygen (FiO2) and a flow of 30 litres per minute (L/min) then up to 50 L/min over 5 to 10 minutes. Patients are monitored and clinically assessed at least every 24 hours to either discontinue, continue, or escalate respiratory support as recommended by the trial protocol [17].

Patients assigned to control arm (SOT) follow the exact same pathway except for a difference in the mode of oxygen therapy delivery following extubation. Patients receive oxygen via nasal prongs or non-rebreathing masks (not humidified and not heated) as is currently

| | Study Period | | | | | |
|---|---|---|---|---|---|---|
| | **Enrolment** | | **Allocation** | **Post-allocation** | | **Close-out** |
| | **Screening** | **Baseline** | **Randomization** | **Discharge** | **30 Days (+30 days) post-op** | **90 Days (+90 days) post-op** |
| Timepoint** | Prior to Surgical Admission (or after admission if in-house urgent) | Surgery Admission | During or After surgery & prior to extubation | Day of Discharge | 30 days (+30 days) post-op | 90 days (+90 days) post-op |
| **Enrolment** | | | | | | |
| Inclusion/Exclusion Criteria | X | | | | | |
| Informed Consent | | X | | | | |
| Demographics | | X | | | | |
| Past Medical History | | X | | | | |
| EuroSCORE II & ARISCAT Risk Assessments | | X | | | | |
| **Interventions** | | | | | | |
| Initiation of high flow nasal therapy | | | X | | | |
| Initiation of standard oxygen therapy | | | X | | | |
| **Assessments** | | | | | | |
| EQ-5D-5L & Barthel Questionnaires | | X | | X | X | X |
| Participant & Family Resource Use Questionnaires | | X | | X | X | X |
| Adverse & Serious Adverse Events Assessed (from the point of extubation | | | X--------------------------------------------------------------X | | | |
| Inpatient Medication Log (to start from the point of extubation) | | | X----------------------X | | | |
| Inpatient Location Log (to start from the point of extubation) | | | X----------------------X | | | |
| Inpatient Oxygen Therapy Log (to start from the point of extubation) | | | X----------------------X | | | |
| Participant Location and Medication Diary | | | | X--------------------------------------X | | |
| ROX Index | | | X----------------------X | | | |
| Record of Respiratory Support Escalation | | | X----------------------X | | | |
| Record of Post-operative Complications | | | X----------------------X | | | |
| Record of Intensive care Length of stay and Re-admissions | | | | X | | |
| Record of Hospital Discharge Destination | | | | X | | |
| Record of Hospital Length of Stay | | | | X | | |

**Fig 1. Study components in accordance with SPIRIT guidelines (EuroSCORE II: European System for Cardiac Operative Risk Evaluation, ARISCAT risk assessment: Assess Respiratory Risk in Surgical Patients in Catalonia, EQ-5D-5L: EuroQol 5-Dimensions 5-Levels, ROX index: Ratio of Oxygen Saturation to FiO2 and Respiratory Rate).**

practiced. Patients receiving SOT are started on 30–40% inspired oxygen at a flow of 2 to 6 L/ min, monitored and assessed as recommended [17].

## Within trial economic evaluation

The economic analysis is designed to inform decisions in the UK, AUS, and NZ. Given the difference between these health systems, the perspective most relevant to decision-making country will be chosen first for analysis. This includes the National Health Service (NHS) and Personal Social Services (PSS) perspective in the UK, the health care system (Medicare/Australian Government Department of Social Services, including costs accruing both in public and private hospitals) perspective in AUS and the health sector and individual perspective in NZ [20–22]. Trial-wide, pooled cost-effectiveness results, if determined following analyses, will use a health and social care sector perspective. The time horizon for the analysis is 90 days post-surgery.

Data collection takes place at five main time points; baseline (surgery admission), randomisation (during or after surgery, prior to extubation), on day of discharge, and on days 30 (+-30-day follow-up window) and 90 (+90-day follow-up window) following surgery. The first patient was recruited on 7/10/2020, the last patient recruited on 19/06/2024 and final follow-up is expected to be completed by 16/12/2024. Some data is also collected between time points, for example inpatient location and patient location diary during follow-up. Data collection records patient outcomes, resources and services utilised from both the health/social care sector and patient/family perspective. Data collection tools were designed in the UK following inputs from the trial team, two consecutive interviews with a patient and their carer, and members of the public. The tools were adapted for use in AUS and NZ with advice from the trial team. The data collection tools account for good practice and methods guidance [20, 23] and uses country specific questions or response categories where necessary. Patient and public involvement and differences in data collection tools for AUS and NZ are reported in S2 Table in S1 Appendix.

## Costing: Identification, measurement, and valuation of resources

**a) Identification of resource use.** Resource use relates to setting up and delivery of HFNT (including the HFNT device, level of use, associated oxygen consumption, training, and maintenance) or SOT (oxygen flow rate, delivery using nasal mask or prong) and its consequences i.e. use of hospital, primary, community and social care until 90 days after surgery for both trial arms. Resource use from the individual/family perspective include expenses for medical, non-medical and social care support services, and time lost by patients and caregivers following randomisation. Table 1 provides a summary of all resources and services identified and collected alongside the trial. As an incremental approach is used in costing, resources common to both arms and those related to conducting the research are excluded.

**b) Measurement of resource use.** Health and social care resources are collected using a variety of patient-specific case report forms (CRFs). Some are completed by research nurses abstracting data from patient-specific records (e.g. oxygen therapy, in-patient location, discharge details), some through research nurses interviewing patients face-to-face or using online, telephone or paper questions as preferred (e.g. quality of life, costs over hospital stay, service use and expenditure post-surgery). Participant location and medication diary following discharge is the only record directly filled in by the patients. Relevant resources used to provide the service but not limited to specific patients (for example, HFNT training, equipment and maintenance) are measured using research records if available. The source and level of resource use measurements taken are provided in Table 1.

**Table 1. Identification and measurement of costs.**

| Resource Use | Description of resources used | Unit of measure | Source, level data collected |
|---|---|---|---|
| **Health & social care sector perspective** | | | |
| HFNT intervention setup | • HFNT device<br>• Training<br>• Maintenance | • Unit cost<br>• Number of trainings<br>• Unit cost | • Published/local sources<br>• Project records<br>• Published/local sources |
| *Initial admission* | | | |
| HFNT | • Amount of oxygen used<br>• Consumables | • FiO2, flow rate<br>• Duration | • CRF, patient specific<br>• Published manual |
| SOT | • Amount of oxygen used<br>• Consumables | • Flow rate<br>• Duration | • CRF, patient specific<br>• Clinical expert opinion |
| Other ventilation support | • CPAP, mechanical ventilation, etc. | • Duration | • CRF, patient specific |
| Inpatient admission, by ward | • Admission to wards, high dependency unit, intensive care unit, etc. | • Duration | • CRF, patient specific |
| Complications after surgery | • Return to operation theatre after initial surgery | • Frequency | • CRF, patient specific |
| Medications | • All medications during admission | • Number of drugs | • CRF, patient specific |
| Diagnostics | • Blood tests, x-ray, ECG, CT scan, microbiology, etc. | • Number of tests | • CRF, patient specific |
| Health professional visits | • Physiotherapy, occupational therapy, social worker, etc. | • Number of visits | • CRF, patient specific |
| Transport—Ambulance use | • Travel to hospital | • Number of visits | • CRF, patient specific |
| *Discharge to day 90 after initial surgery* | | | |
| Transport -Ambulance use | • Travel to hospital | • Number of visits | • CRF, patient specific |
| Hospital admission | • Overnight admission to hospital, nursing home | • Number of visits | • CRF/GP records, patient specific |
| Medications | • All medications from discharge to day 90 | • Number of drugs | • CRF, patient specific |
| Diagnostics | • Blood tests, x-ray, etc. | • Number of tests | • CRF, patient specific |
| Day hospital services | • Outpatient clinics, physiotherapy, accident & emergency, etc. | • Number of visits | • CRF, patient specific |
| GP surgery services | • GP surgery to see a doctor, nurse, other professionals | • Number of visits | • CRF, patient specific |
| Home visits/ telephone consultation | • Home visit from health or social care professionals<br>• Telephone consultation | • Number of visits<br>• Number of consultations | • CRF, patient specific |
| Social support services | • Care worker, overnight care, meals on wheels, etc. | • Number of contacts | • CRF, patient specific |
| **Patient/family perspective** | | | |
| *Initial admission* | | | |
| Admission | • Contribution/payment during initial surgery admission | • Patient reported cost | • CRF, patient specific |
| Transport | • Travel to and from hospital for surgery.<br>• Visitor travel | • Patient reported travel time | • CRF, patient specific<br>• CRF, patient specific |
| Miscellaneous | • Food, drink, accommodation expense | • Patient reported cost | • CRF, patient specific |
| Time lost to illness | • Main activity done by patient if not in surgery. For example, paid work, unpaid work, caring etc. | • Time lost | • CRF, patient specific |
| Accompanying person time lost due to caring | • Main activity done by accompanying person if not caring | • Time lost | • CRF, patient specific |
| *Discharge to day 90 after initial surgery* | | | |
| Admission | • Contribution/payment for nursing home, residential care, etc. | • Patient reported cost | • CRF, patient specific |
| Diagnostics | • Contribution/payment for blood tests, x-ray, etc. | • Patient reported cost | • CRF, patient specific |
| Social support services | • Contribution/payment for home help, night care, meals on wheels, etc. | • Patient reported cost | • CRF, patient specific |
| Medical supplies and equipment | • Medical supplies: For example, prescriptions, aspirin, compression socks, nebulizer, etc.<br>• Equipment: For example, walking aids, special bedding, etc. | • Patient reported cost<br>• Patient reported cost | • CRF, patient specific<br>• CRF, patient specific |
| Transport | • Travel to hospital, GP surgery, etc. | • Patient reported travel time | • CRF, patient specific |
| Home adaptation | • Adaptation made to home. For example, ramp, stair-lift, etc. | • Patient reported cost | • CRF, patient specific |
| Time lost to illness | • Care arrangements, time off work, etc. | • Patient reported time lost | • CRF, patient specific |

*(Continued)*

**Table 1.** (Continued)

| Resource Use | Description of resources used | Unit of measure | Source, level data collected |
|---|---|---|---|
| Accompanying person time lost due to caring | • Main activity done by accompanying person if not caring | • Time lost | • CRF, patient specific |

(HFNT–High Flow Nasal Therapy, FiO2—fraction of inspired oxygen, CRF–Case Report Form, SOT–Standard Oxygen therapy, CPAP—Continuous positive airway pressure, ECG—Electrocardiogram, CT-Scan—Computed Tomography Scan, GP—General practitioner)

The amount of oxygen used with HFNT and SOT forms a relevant differential resource between the trial arms and hence is costed. As the central oxygen supply is common to both HFNT and SOT, the analysis measures only the variation in amount of oxygen supplied either by HFNT or SOT at a patient-specific level. For SOT, the CRF records oxygen flow rate titration in L/min and duration of use to give total quantity of oxygen used. For HFNT, the CRF records FiO2 and the gas flow rate (L/min). Using an indicative chart, patient-specific data on FiO2 and gas flow rate will be used to determine amount of oxygen used by HFNT [24]. Consumables (e.g., oxygen interface, tubing, nasal prong) will be quantified using patient-specific duration of oxygen therapy along with available duration of use guidance given by manufacturer or using clinical expert opinion [25].

The changes to the data collection instruments in AUS and NZ included contextual country-specific questions, responses and wording revisions for resource use data collection. This mainly included replacing country-specific questions on ethnicity, education, healthcare concession and out-of-pocket payments for ambulance travel, health professional visit and telephone consultations in AUS and NZ. Additionally, during the trial, amendments were made to data collection tools to improve data completion. Further details on data collection amendments are available in S3 Table in S1 Appendix.

**c) Valuation of resource use.** All health and social sector resources will be valued using national costs (for example, national cost collection in the UK) for the latest available year [26–32]. Any unit cost not in the corresponding year will be adjusted using a health- or country-specific inflation index (for example, NHS cost inflation index published in Personal Social Services Research Unit (PSSRU) cost manual) [31, 33–36]. If national unit costs are unavailable, published literature or local cost sources will be used. Capital costs will be annualised using country-specific discounting recommendations (3.5% UK, 5% AUS, 3.5% NZ) or using the 3.5% UK recommendation for pooled results, further assessed in sensitivity analysis [20, 21, 37]. For example, in base-case, the HFNT device will be annualised using an average lifespan of 5 years, 3.5% discount rate and an annualised maintenance cost using available sources. Patient reported expenses will be used to value patient perspective costs. Time lost by patients and caregivers will be valued using the human capital approach based on gross national average wages [38–40]. Unpaid work will be valued using wage earned for similar work on a paid basis or using national average earnings if unavailable.

**d) Computation of total costs.** Total cost will be computed at a patient level by summing the components contributing to the cost of intervention delivery, initial admission and up to 90 days post-surgery relevant to health/social care sector and patient/family perspectives. For example, in base-case, per patient cost of HFNT arm in UK will include NHS/PSS perspective costs for HFNT delivery, initial admission and service utilisation until end of follow-up respectively. Total costs will be obtained by multiplying the quantity of resource use by unit cost across each subcomponent and then summing across. Some costs (for example, HFNT device)

are expected to be shared between patients. In such cases, annualization, life-span and apportioning rules (for example, number of patients using the device) will be used as appropriate.

## Outcomes: Identification, measurement, and valuation of health outcomes

**a) Identification of patient reported outcomes.** Days alive and at home after surgery (DAH90), a validated patient-reported metric, is the primary outcome endpoint for NOTACS trial [41]. DAH90/DAH30 define home as a person's usual abode at 90 (and 30) days post-surgery. Escalation of care or time away from usual abode during follow-up period will be counted as a day away from home as per defined protocol. DAH is sensitive to changes in surgical risk and is considered a superior measure of quality of surgery and care than standard complication and mortality rates [42, 43]. The CRF records changes in the participant's overnight living location throughout follow-up. It includes details on whether stay was planned and if support for daily living activities was needed. The DAH measure will be used in cost-effectiveness analysis (CEA) as it was identified as the most important primary outcome by patient and public representatives during development of the proposal.

For cost-utility analysis (CUA), a standardised generic health related quality of life measure, the EQ-5D-5L, relevant to the UK, AUS, and NZ is used [20–22, 44]. The EQ-5D-5L consists of two parts; the first asks respondents to categorise their health today on five dimensions (mobility, self-care, usual activities, pain/discomfort, anxiety/depression) and five response levels (no problems, slight problems, moderate problems, severe problems, unable to) for each dimension, and the second part asks respondents to rate perception of their own overall health today on a scale of 0 (worst) to 100 (best) imaginable health.

**b) Measurement and valuation of patient reported outcomes.** DAH90 and DAH30 are collected using a mix of CRFs completed by patients and research nurses from date of index surgery to 90 days after surgery. Patients are given the location diary on discharge and have the option of completing follow-up questionnaires on paper, online or via telephone. Patients not completing follow-up CRFs at 30 and 90 days are contacted regularly up to 30+30 and 90 +90 days as per trial protocol. Information may be gathered from the patient's general practitioner (GP) surgery in case of non-response or to verify patient responses as per consent.

DAH90/DAH30 valuation accounts for any increase in care level from patient's previous residence. For example, hospital readmission or additional care requirement at 'home' within 90 days of surgery is subtracted from the total number of days at home. Staying away from 'home' for social reasons without any additional care or support is considered as home. The protocol for what counts as days away from home is predefined and available in the update of the published statistical analysis plan (submitted and under review). The location is based on where a patient spends the night. Patient records reporting baseline or follow-up abode as 'Other' will be reviewed for whether the 'other' should be classed as 'home' and agreed on a case-by-case basis by at least two blinded members. An additional blinded member will be consulted if consensus is not reached. If a patient living at home is discharged from the hospital on day 6 after surgery but is subsequently readmitted for 4 days and then returns home until 90 days post-surgery, they would be assigned 80 DAH90.

The base case statistical analysis for the trial follows the approach used by Myles et al., to count days at home i.e. when a patient dies—a DAH value of 0 occurs for the whole period [18, 41]. For example, if a patient dies while still in the hospital or at home before 90 days, they are assigned 0 DAH90. This means no value is attributed whether a person lives for 1 or 89 days within the study period and no change can be seen if survival changes over time. The trial is not designed to capture change in survival and one reasoning to accept such a scoring is that it takes a more conservative view of benefit. However, for the economic evaluation there are

two problems in using this approach. First there is evidence that costs and outcomes can be related, and higher costs are particularly related to proximity to death [45–47]. Through an increase in outliers, and potential miscasting of the relationship between costs and effects, assigning a zero value to all people dying could significantly bias regression results for cost-effectiveness. Second the cost-utility analysis, part of the suite of tools in economic evaluation used to support NICE's decision-making, does account for all days lived. Therefore, the base-case cost-effectiveness analysis will be conducted using all recorded days at home and dubbed 'DAH90(HE)' to represent a difference with calculation of the main trial outcome DAH90, which will be used in sensitivity analysis, to allow comparison with the main trial analysis.

Responses recorded on the EQ-5D-5L tool will be converted into a health state utility score using UK, AUS and NZ specific value sets [48–50]. As recommended in the UK, the EQ-5D-5L data will be mapped onto the 3L value set [48, 51], with QALYs calculated using the area under curve method [52]. The study design offers patients a tolerance window of 90 days for 90-day follow-up (and 30+30 days) to complete data collection. Because utility values may be expected to rise over time post-surgery, sensitivity analysis will explore the impact of using linear interpolations to obtain utility values at 90 days [53].

## Analysis

**a) Handling missing data.** Before commencing analysis, data quality checks will be conducted and any duplicates, missing or extreme values will be identified and documented. Any missing data will be cross-checked against the CRF and, where necessary, patient records reviewed for procedural error. Clear outliers, such as those exceeding 4 standard deviations from the centre of the distribution, will be investigated further and may be excluded or treated as missing values. However, if a series of potential outliers appears within a skewed distribution, they may be retained. We will examine the data characteristics to select candidate distribution and assess model fit using information criteria, to ensure the best representation of the data.

The pattern and amount of missing data at each time point will be summarised using frequency and percentage plots, with pattern of missingness for resource use and outcomes. We will use logistic regression to investigate if missingness can be predicted by baseline or observed outcomes at each data collection point [54]. If none of the missing variables are predicted by other variables, a missing completely at random (MCAR) mechanism will be assumed and if the percentage of missing data is below 5% for resource use and outcomes of the primary analysis, a complete case analysis (CCA) will be undertaken. This is unlikely to be the case due to the need to aggregate resource use and costs across many variables. In this case, a missing at random (MAR) mechanism will be assumed and multiple imputation (MI) methods will be used. Sensitivity analysis will assess the impact of using a missing not at random (MNAR) mechanism using pattern mixture model approach [55, 56]. We will use multiple imputation chained equations (MICE) with predictive mean matching, stratified by treatment arm and country to estimate missing cost and outcome data for base-case. We will explore censoring as a special case of missing data, particularly right censoring from loss to follow-up to judge the impact on estimating mean values for costs, DAH90 and QALYs using an inverse probability weighting approach [54]. Imputation models will be determined during analysis as it is important that the model includes variables that are good predictors of missing variables. We will consider variables such as age, sex, baseline Barthel and/or EQ-5D-5L scores, baseline health service use, BMI, ethnicity, number of co-morbidities, whether surgery is elective, Euro Score II, ASA, and ARISCAT score. The number of imputed data sets will be higher than the percentage of patients with missing data in the subset of variables included in the missing data

mechanism tests to achieve stable imputation [56, 57]. For each imputed dataset, ten imputation cycles will be used. The imputation results will be validated by checking the distribution of imputed values against observed values. Rubin's rule will be used to generate a complete single data set for analysis, encompassing the variation and uncertainty preserved in the multiple imputed data sets [55]. Total costs, DAH90(HE), utility index score at day 90 and total QALYs with CCA, pre-imputed and imputed results will be presented for HFNT versus SOT comparison.

**b) Statistical analysis of differences in resource use, costs, and outcomes for cost-effectiveness and cost-utility analyses.** The intention-to-treat principle will be followed for base-case analyses. Resource use, costs and outcomes will be summarised by trial arm; a) at discharge, 30- and 90-days, and b) by country at 90-days using descriptive statistics (frequency, mean (standard deviation), median, minimum (including % of zeros), maximum, IQR, 95% confidence interval around the mean, histograms and box plots) along with tests of normality. Significant differences at 5% level, using raw and imputed data, will be tested using two-sample t-tests of equality for normally distributed variables, and if assumptions of normality and homoscedasticity are not met, non-parametric Mann-Whitney-Wilcoxon test and boot-strapped t-tests will be used. This analysis will be repeated for a complete case analysis if feasible. Changes in the EQ-5D-5L health profile will be summarised using the Paretian Classification of Health Change and by reporting the ten most frequently observed health states at baseline and day 90 for both trial arms [58]. A Health Profile Grid will be developed for visual representation [58].

Regression models will be fitted for mean costs and QALYs, with and without imputed values. Models will include intervention arm and variables to adjust for baseline patient characteristics (age, sex, Barthel, EQ-5D score, health service use, BMI, ethnicity, number of co-morbidities and whether surgery is elective, Euro Score II risk score, ASA and ARISCAT score). We will however be unable to control for baseline differences in patient/family borne costs. We will examine data prior to selecting a regression model but will consider generalised linear and mixed models given their ability to handle skewness and large numbers of zeros and will test log-link and more flexible forms to examine goodness of fit. We will assess the relationship between baseline covariates for collinearity in the decision for covariate inclusion in the final model. We will also examine the impact of accounting for the correlation between costs and QALYs, by using seemingly unrelated regression analysis with and without correcting for confounding at baseline [59].

The difference in costs between the trial arms will be divided by the difference in outcomes between the trial arms, to give incremental cost-effectiveness/utility ratios. Incremental cost-utility ratios (ICURs) will be compared with country-specific costs-effectiveness thresholds (CET) used in the UK (£20,000/QALY), AUS (AU$ 28,033/QALY) and NZ (NZ$ 40,000/QALY) [20, 60, 61]. Fully pooled results will be reported in the UK currency (country-specific unit cost applied to resource use followed by summing of all costs to UK currency using purchasing power parity (PPP)) and assessed against the CET value for UK. Country-specific incremental net monetary benefit (NMB) statistics, (INMB = incremental benefit * CET– incremental cost) will be presented with 95% CI limits. As no CETs are available for DAH outcome, the incremental cost-effectiveness ratios (ICERs) will be used to support wider decision making for any clearly dominant or dominated strategies. We will follow the recommended framework to present cost-effectiveness and cost-utility estimates for this multinational trial [62, 63] and present following; a) country-specific ICUR estimates using country-specific data on cost and QALY respectively b) country-specific ICUR estimates using trial-wide resource use valued against country-specific unit costs and trial-wide EQ-5D responses valued against domain level EQ-5D value set for chosen country) and, depending on evidence of

**Table 2. Methods and perspective for presenting the cost-effectiveness results for multinational NOTACS trial.**

| Analysis | Primary (perspective) | Method | Secondary (perspective) | Method |
|---|---|---|---|---|
| **a) Country specific** | | | | |
| UK | NHS and PSS | Only UK specific data | NHS and PSS<br>+ Individual (OOPE)<br>+ value of time lost | Only UK specific data |
| AUS | Health care system (includes public/private) | Only AUS specific data | Health care system (includes public/private)<br>+ value of time lost | Only AUS specific data |
| NZ | Health sector<br>+ Individual (OOPE) | Only NZ specific data | Health sector<br>+ Individual (OOPE)<br>+ value of time lost | Only NZ specific data |
| **b) Country specific—trial pooled** | | | | |
| UK | NHS and PSS | Resource use–UK, AUS, NZ combined<br>Unit costs–UK<br>Outcomes–UK, AUS, NZ combined and valued using UK tariff | NHS and PSS<br>+ Individual (OOPE)<br>+ value of time lost | Resource use–UK, AUS, NZ combined<br>Unit costs–UK<br>Outcomes–UK, AUS, NZ combined and valued using UK tariff |
| AUS | Health care system (includes public/private) | Resource use–UK, AUS, NZ combined<br>Unit costs–AUS<br>Outcomes–UK, AUS, NZ combined and valued using AUS tariff | Health care system (includes public/private)<br>+ value of time lost | Resource use–UK, AUS, NZ combined<br>Unit costs–AUS<br>Outcomes–UK, AUS, NZ combined and valued using AUS tariff |
| NZ | Health sector<br>+ Individual (OOPE) | Resource use–UK, AUS, NZ combined<br>Unit costs–NZ<br>Outcomes–UK, AUS, NZ combined and valued using NZ tariff | Health sector<br>+ Individual (OOPE)<br>+ value of time lost | Resource use–UK, AUS, NZ combined<br>Unit costs–NZ<br>Outcomes–UK, AUS, NZ combined and valued using NZ tariff |
| **c) Fully pooled** | | | | |
| Trial | Health and social care sector | Resource use–UK, AUS, NZ<br>Unit costs–UK, AUS, NZ<br>Total cost–UK, AUS, NZ combined to UK currency using PPP<br>Outcomes–UK, AUS, NZ tariffs used on respective data for single combined QALY measure | Health and social care<br>+ Individual (OOPE)<br>+ value of time lost | Resource use–UK, AUS, NZ<br>Unit costs–UK, AUS, NZ<br>Total cost–UK, AUS, NZ combined to UK currency using PPP<br>Outcomes–UK, AUS, NZ tariffs used on respective data for single combined QALY measure |

(PPP–Purchasing power parity, NHS and PSS–National Health Services and Personal and Social Services, OOPE–Out-of-pocket expenditure, AUS–Australia, NZ–New Zealand, UK–United Kingdom)

heterogeneity, c) pooled trial-wide ICUR estimate by applying country-specific unit costs to resource use followed by summing costs to the common UK currency using PPP and summing country-specific QALYs using country-specific value sets into a pooled QALY value. The Likelihood ratio test and range test of qualitative interaction will be used to examine if the direction of cost-effectiveness varies between the countries [64, 65]. The likelihood ratio and standardised range test statistic will be compared against respective critical values at a significance level of $\alpha = 0.05$ for the three groups to indicate heterogeneity [64, 66]. We will explore the use of shrinkage estimates as an alternative to trial-wide pooling of cost-effectiveness results [67]. Table 2 provides the methods and perspective considered in presenting the range of cost-effectiveness results.

**c) Handling uncertainty.** Parameter uncertainty will be handled using one-way sensitivity analysis (OWSA) and scenario analysis to assess the impact of following changes, including but not limited to:

i. Costs

- HFNT intervention setup costs, including assumptions about length of life, device use and price.

- Oxygen therapy cost, any variation in oxygen consumption calculation.

- Any significant assumption in measurement or valuation methods for costs, including outliers (for example, methods for follow-up/intravenous medication data collection, discount rate, etc).

- Cost at 30 days.

- Perspective (impact of including patient/family costs)

ii. Outcomes

- DAH90 to assess the impact of assigning a DAH value of zero for those dying between zero to 90 days post-surgery.

- Using linear interpolate and 90+90th day value as a proxy for 90th day utility value in QALY calculation.

- Outcomes at 30 days.

 Sampling uncertainty will be characterised using 95% CI limits for ICUR/ICER values based on non-parametric bootstrapping along with cost-effectiveness plane (CEP) and cost-effectiveness acceptability curves (CEAC) to inform decisions [20].

 Heterogeneity will be assessed using sub-group analysis to improve robustness of our findings. Two potential sub-groups will be assessed: a) patients requiring escalation of oxygen therapy, as it may indicate treatment failure driving higher resource utilization and previous evidence suggests that HFNT may benefit patient subgroups receiving complete protective ventilation [68]. b) patients with differential time on treatment (at least 8 or 24 hours) due to the potential impact on costs (for example, amount of oxygen used) and differential outcomes.

## Discussion

The NOTACS trial is the first international evidence comparing HFNT versus SOT in adult patients at high risk of developing PPC undergoing cardiac surgery. The trial addresses clinical and economic evidence gaps regarding the use of HFNT [10, 11]. A recent pilot study in this same population subgroup saw a reduction in length of hospital stay by 29% and re-admission to ICU dropped from 14% to 2% with HFNT compared to SOT [69]. HFNT can potentially lead to significant cost-savings and offer good value for investment. The NOTACS trial aims to inform policy recommendations in the UK, AUS, and NZ.

### Strengths

This study has several strengths. First, this is the first study to measure and account for the cost of oxygen therapy with HFNT or SOT following cardiac surgery using prospective, patient-level data. Second, the analysis takes a multinational approach, incorporating methods and guidance relevant to decision-making in three diverse healthcare systems. Our analysis aims at maximising the balance between using available data from the trial with the perspective and methods most relevant to each country. Third, data collection was influenced by inputs from patients and the public, is country-specific, and records both patient and family costs, and time lost at a patient-level. This yields a potentially valuable source of evidence for broader understanding of the economic implications of HFNT intervention. Fourth, the range of estimates planned for, along with a wide range of sensitivity analyses will give a good indication of the robustness of findings. Finally, the findings can indicate resource utilisation patterns and

health benefits in other sub-populations (e.g. following abdominal surgery) where HFNT can be used.

## Limitations

There are limitations to the planned analyses. Firstly, in presenting the pooled cost-effectiveness results, the study adopts a health and social care sector perspective. This includes the costs of community-based services and home or continuing care, as recommended in the UK, AUS and NZ [20, 70, 71]. However, this pooled analysis will not include patient perspective costs, which form an important consideration in AUS and NZ. This exclusion may limit the applicability of the pooled trial wide results to these countries. The decision on perspective for pooled analysis was made because the trial was initially designed for and predominantly recruits from the UK. We do however plan to present results ranging from country-specific results accounting for patient perspective costs to pooled trial wide results, presenting relevant economic evidence for decisions. Moreover, the procedure of fully pooling the results can identify and provide empirical evidence of homogeneity or heterogeneity across these countries. Secondly, the cost-effectiveness analysis findings using DAH outcome measure relies on participation diaries completed by patients to record any changes in location. Despite efforts to follow-up with patients' GP practice for incomplete records, there still may be instances where this information is unavailable, potentially influencing the cost-effectiveness findings. Thirdly, the analysis focuses on the cost-effectiveness of HFNT for patients included in controlled trial settings which may differ from the broader population or more real-world clinical settings. Finally, results are limited to 90 days of follow up and cost-effectiveness over a longer period may differ.

## Supporting information

**S1 Appendix.**
(DOCX)

**S2 Appendix.**
(DOCX)

**S1 Protocol.**
(PDF)

## Acknowledgments

We wish to thank all patients, family members, and staff members from all the participating sites for their involvement in this study.

## Author Contributions

**Conceptualization:** Siddesh Shetty, Richard Norman, Jacquita Affandi, Sarah Dawson, Julia Fox-Rushby.

**Funding acquisition:** Julia Fox-Rushby.

**Methodology:** Siddesh Shetty, Melissa Duckworth, Richard Norman, Jacquita Affandi, Julia Fox-Rushby.

**Project administration:** Melissa Duckworth.

**Software:** Siddesh Shetty, Sarah Dawson, Julia Fox-Rushby.

**Supervision:** Julia Fox-Rushby.

**Visualization:** Siddesh Shetty.

**Writing – original draft:** Siddesh Shetty.

**Writing – review & editing:** Siddesh Shetty, Melissa Duckworth, Richard Norman, Jacquita Affandi, Sarah Dawson, Julia Fox-Rushby.

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
