## [Decision Letter · Decision Letter 0]

6 Nov 2024

PONE-D-24-41068The international trial of nasal oxygen therapy after cardiac surgery (NOTACS) in patients at high risk of postoperative pulmonary complications: Economic evaluation protocol and analysis planPLOS ONE

Dear Dr. Fox-Rushby,

Thank you for submitting your manuscript to PLOS ONE. After careful consideration, we feel that it has merit but does not fully meet PLOS ONE’s publication criteria as it currently stands. Therefore, we invite you to submit a revised version of the manuscript that addresses the points raised during the review process.

We look forward to receiving your revised manuscript.

Kind regards,

Giuseppe Pipitone, M.D.

Academic Editor

PLOS ONE

4. One of the noted authors is a group or consortium [NOTACS investigators, NOTACS Trial Steering Committee and NOTACS DMEC Committee]. In addition to naming the author group, please list the individual authors and affiliations within this group in the acknowledgments section of your manuscript. Please also indicate clearly a lead author for this group along with a contact email address.

6. We note that there is identifying data in the Supporting Information file <S2 Appendix PLOS ONE_DMEC.docx>. Due to the inclusion of these potentially identifying data, we have removed this file from your file inventory. Prior to sharing human research participant data, authors should consult with an ethics committee to ensure data are shared in accordance with participant consent and all applicable local laws.

-Location data

Please remove or anonymize all personal information, ensure that the data shared are in accordance with participant consent, and re-upload a fully anonymized data set. Please note that spreadsheet columns with personal information must be removed and not hidden as all hidden columns will appear in the published file.

7. We note that the original protocol that you have uploaded as a Supporting Information file contains an institutional logo. As this logo is likely copyrighted, we ask that you please remove it from this file and upload an updated version upon resubmission.

Additional Editor Comments:

Dear Corresponding Author,

Please carefully read reviewer's comments and give us a point by point response in order for me to take a final decision.

Kind regards

Giuseppe Pipitone

Reviewers' comments:

Reviewer's Responses to Questions

**Comments to the Author**

1. Does the manuscript provide a valid rationale for the proposed study, with clearly identified and justified research questions?

Reviewer #1: Yes

Reviewer #2: Yes

2. Is the protocol technically sound and planned in a manner that will lead to a meaningful outcome and allow testing the stated hypotheses?

Reviewer #1: Yes

Reviewer #2: Yes

3. Is the methodology feasible and described in sufficient detail to allow the work to be replicable?

Reviewer #1: Yes

Reviewer #2: Yes

4. Have the authors described where all data underlying the findings will be made available when the study is complete?

Reviewer #1: Yes

Reviewer #2: Yes

5. Is the manuscript presented in an intelligible fashion and written in standard English?

Reviewer #1: Yes

Reviewer #2: Yes

6. Review Comments to the Author

You may also provide optional suggestions and comments to authors that they might find helpful in planning their study.

Reviewer #1: The study addresses an interesting topic. The description of each phase is clear and detailed.

I have a few remarks on the data analysis only. In details:

1. The definition of an outlier is always with respect to a specific distribution. A value that might be atypical with respect to, e.g., the Poisson distribution may be not atypical for the Geometric, the Negative Binomial or the Conway Maxwell Poisson distributions, just to mention the most widely used count data distribution (along with their zero-inflated counterparts). Similarly, I am strongly against the transformation of the data. It is not the data that should adapt to a specific model, but rather it is important to find the most appropriate model for the data at hand; log-transforming the data leads to a different shape and scale.

2. Similarly, I do not really get how you can say that a missing value is MAR, MCAR or MNAR. Sensitivity analysis is often considered, under different missing data mechanisms (please refer to https://www.routledge.com/Handbook-of-Missing-Data-Methodology/Molenberghs-Fitzmaurice-Kenward-Tsiatis-Verbeke/p/book/9780367739294 for further details). Thus, I strongly suggest to discuss more details on the way missing data will be treated, allowing for NMAR mechanisms. The authors state that multiple imputation is used, but it is not clear how many datasets will be considered and how the imputed datasets will be checked, etc.

3. I see that parametric t-tests will be considered. Be aware that parametric tests must met quite strong assumptions to lead to reliable inference (e.g. Gaussianity, homoscedasticity, etc.).

4. Regression models will be employed, but which models? Generalized linear models, generalized additive models, their mixed-effects counterparts? Survival models could also be employed. Please, provide more details, in particular of the way zero-inflation will be addressed (see e.g. Winkelmann, R. (2004). Health care reform and the number of doctor visits—an econometric analysis. Journal of Applied Econometrics, 19(4), 455-472; Alfò, M., & Maruotti, A. (2010). Two‐part regression models for longitudinal zero‐inflated count data. Canadian Journal of Statistics, 38(2), 197-216; Green, J. A. (2021). Too many zeros and/or highly skewed? A tutorial on modelling health behaviour as count data with Poisson and negative binomial regression. Health Psychology and Behavioral Medicine, 9(1), 436-455.)

Reviewer #2: Review of “The international trial of nasal oxygen therapy after cardiac surgery (NOTACS) in

patients at high risk of postoperative pulmonary complications: Economic evaluation

protocol and analysis plan” with manuscript number PONE-D-24-41068

This is a protocol paper describing the cost collection, valuation of resources, analysis methods and sensitivity analyses for an international trial for patients at high risk of postoperative pulmonary complications. The authors have submitted a well-written and highly detailed protocol paper. The planned analysis includes all the relevant components to a satisfactory economic evaluation utilizing cost-effectiveness and cost-utility analysis. However, there are some points I would like to raise:

Overall comments

Why choose to pool the results of the country-specific data? This will not inform any specific decision maker. It is better to use a base-case country for the main analysis and then conduct sensitivity analyses for the remaining countries. The results from the pooling analysis will not make sense to any decision maker and just seems rather non-important because the authors are already conducting the important country-specific analyses.

Why is the resource use for the family/friends not measured in time lost, as it is for the patient, and rather measured in time spent? The time spent with the patient is time lost at, for example, work? How do you value “time spent”?

I would like a more thorough description of the sub-group analysis – right now it seems forced and though out at the last minute. Why are these two sub groups chosen? Could there be other relevant sub groups that could be used in analysis? How about those that previously have had the same condition compared to the ones that are first-timers? How about social inequality aspects? We know that outcomes differ between patients with different socio-economic levels.

I would like a more comprehensive explanation of how the cost collection has been influenced by patients and the public. What does this mean, how was it done in detail, how did it differ between the countries and how does this affect the results?

What is SOT? The first time this is mentioned in the Introduction it is mentioned using only the abbreviation and not explained beforehand. It is important to know what the comparison arm is and what is consists of.

7. PLOS authors have the option to publish the peer review history of their article (what does this mean?). If published, this will include your full peer review and any attached files.

Reviewer #1: No

Reviewer #2: No

---

## [Author Response · Author response to Decision Letter 0]

17 Dec 2024

Response to reviewers has been submitted as a separate word file in the submission system

---

## [Editor Report · Decision Letter 1]

15 Jan 2025

The international trial of nasal oxygen therapy after cardiac surgery (NOTACS) in patients at high risk of postoperative pulmonary complications: Economic evaluation protocol and analysis plan

PONE-D-24-41068R1

Dear Authors,

We’re pleased to inform you that your manuscript has been judged scientifically suitable for publication and will be formally accepted for publication once it meets all outstanding technical requirements.

Kind regards,

Giuseppe Pipitone, M.D.

Academic Editor

PLOS ONE
---

## [Editor Report · Acceptance letter]

17 Jan 2025

PONE-D-24-41068R1 

PLOS ONE

Dear Dr. Fox-Rushby, 

I'm pleased to inform you that your manuscript has been deemed suitable for publication in PLOS ONE. Congratulations! Your manuscript is now being handed over to our production team.

Kind regards, 

on behalf of

Dr. Giuseppe Pipitone 

Academic Editor

PLOS ONE